# Factors Affecting Inpatients’ Mortality through Intentional Self-Harm at In-Hospitals in South Korea

**DOI:** 10.3390/ijerph20043095

**Published:** 2023-02-10

**Authors:** Sulki Choi, Sangmi Kim, Hyunsook Lee

**Affiliations:** 1College of Applied Health Science, Biomedical Health Information Science, University of Illinois at Chicago, 1919 W Taylor St, Chicago, IL 60612, USA; 2Department of Health Management, Jeonju University, 303 Cheonjam-ro, Wansan-gu, Jeonju-si 55069, Republic of Korea; 3Department of Health Administration, Kongju National University, 56 Gongjudaehak-ro, Singwan-dong, Gongju-si 32588, Republic of Korea

**Keywords:** intentional self-harm, in-hospital suicide, mortality, South Korea

## Abstract

This study aimed to identify the patient characteristics, comorbidities, risk factors, and means of the self-harm of patients who attempt self-harm in and outside of a hospital, and to determine the characteristics of death by suicide among survival and death patient groups in South Korea. This study used data from the Korean National Hospital Discharge In-depth Injury Survey conducted from 2007 to 2019. In total, 7192 outpatient participants and 43 inpatient participants performed self-harm. Frequency analysis, chi-square tests, Fisher’s exact test, and logistic regression analysis were performed using STATA, version 15.0 (StataCorp), and statistical significance was set at 5%. Thirty-one inpatients who performed self-harm survived, and 12 died. Among male inpatients, the older they were, the higher the rates of self-harm and mortality rates due to falls and poisoning if they had comorbidities and financial problems. In addition, the rate of self-harm attempts within a short period after hospitalization was high. Our evidence of the characteristics of patients who performed self-harm in the hospital and the influencing factors of self-harm can be used as primary data for predicting patients at a high risk of self-harm and for creating preventative policies to reduce the risk of self-harm among inpatients in South Korea.

## 1. Introduction

In healthcare, patient safety is a global priority [1], and there is a substantial body of research on patient safety in general hospital services [2]. Patient safety can be maintained and accidental damages in healthcare prevented by reducing the chances of medical errors or accidents and avoiding exposing patients to risks. In 1999, after the “To Err is Human: Building a Safer Health System” report of the American Academy of Medicine was published, patient safety awareness received considerable attention in the healthcare field [3]; according to this report, 2.9–3.7% of inpatients experienced some form of adverse event, and 6.6–13.6% of them died. Furthermore, in the United States, the predominant paradigm of patient safety (Patient Safety paradigm I) focuses on “unintended or unexpected incidents that could have or did lead to harm for one or more patients receiving NHS funded-healthcare” [4].

Regarding suicide, the World Health Organization (WHO) defines it as a “case of death by the fatal results caused by suicidal behavior” and suicidal behavior as “self-injurious behavior while recognizing intentions and motives of death” [5]. Thus, such thoughts are considered a continuum of concepts related to suicidal attempts or behaviors that indicate suicidal risks and warnings. Suicide is among various patient safety-related accidents that occur in medical institutions. The news of suicide in hospitals spreads quickly; patients who hear it can easily fall into depression and anxiety, even if the person who committed suicide was unknown to them, due to death being a sensitive matter [6]. It has also been reported that medical staff who experience hospital suicide feel shocked, fearful, and guilty, increasing their distrust of hospital management. Moreover, hospital suicides result in a psychological and institutional breakdown regarding the stakeholders’ typical expectations about patient safety [7].

In many countries, there is no reliable estimation of the numbers related to inpatient suicide. Based on the reports of the National Violent Death Reporting System during 2014–2015 on hospital inpatient suicide, 73.9% occurred during psychiatric treatment, and it is estimated that between 48.5% and 64.9% of hospital inpatient suicides occur per year in the United States of America [8]. Some behaviors related to suicide or suicide attempts in hospital settings include jumping, cutting, hanging, poisoning, drowning, dialysis, head trauma, swallowing objects, and self-hitting on the wall [9]. Among patients, hanging is the most common method of suicide [10]. In Hubei province, China, patients from 48 general hospitals died by suicide within a week of diagnosis (i.e., for either malignant tumors or chronic diseases). It was reported that this was due to their economic status, as they had low incomes, chronic diseases, low education, were engaged in agriculture, and were economically inactive [11].

Regardless of these numbers, medical institutions must be safe places for all involved parties, including patients, their parents, clinicians, and administrators. We hope the results can be used as fundamental data for developing programs that complement and strengthen policies and monitoring systems related to patient safety in medical institutions. They can also be used for developing appropriate measures to reduce the number of hospital suicides.

## 2. Method

### 2.1. Data

Since July 2016, following the enactment of the Act of Patient Safety in South Korea, patient safety accidents occurring at medical institutions have been reported through the Korea Patient Safety Reporting & Learning System. Fifty-three cases of hospital suicide or self-harm were reported from July 2016 to July 2017, 158 cases were reported in 2019, and 127 cases were reported in the first half of 2022 [12]. This study used data from the Korean National Hospital Discharge In-depth Injury Survey of the Ministry of Health and Welfare and the Korea Centers for Disease Control and Prevention. This national survey-based project has been conducted since 2005. It collects data from general hospitals with more than 100 beds to establish and evaluate chronic disease and injury prevention policies. The data set comprises geographic data; patient visit data; disease and treatment data; data on intentional damage; location, date, and mechanism of injury; activity at the time of injury; types of transport accidents; suicide risk factors; and poisoning substances, and it stems from different medical institutions and the general population. In Korea, the government provides medical insurance and public assistance programs for financially disadvantaged people, including providing financial support to people so they may take care of their primary needs, such as food and clothing. For the sample of this study, 9% of all patients discharged annually from 170 medical institutions were randomly selected, and we investigated data from their medical records.

The inclusion criteria were discharged patients who performed or attempted intentional self-harm (i.e., an action necessary for a person to commit suicide or a suicide attempt) from 2007 to 2019. The data from 2005 and 2006 were excluded because the related data sets differed in the damage mechanisms and activities at the time of injury. Intentional self-harm was defined as cases with the extrinsic codes X60–X84 and Y10–Y34, based on the International Classification of Disease and cause of death [13,14,15]. In total, 7235 participants aged over 9 and under 100 were selected for the final sample.

### 2.2. Variables

The dependent variable was the place where patients performed self-harm, and it was categorized as self-harm in and outside the hospital setting. Meanwhile, survival and death rates were divided according to the treatment outcomes of the patients who attempted self-harm in the hospital setting. The independent variables were patient characteristics, comorbidities, risk factors, means of self-harm, treatment outcomes, and the period from admission to the date of self-harm attempt for all patients who attempted self-harm in the hospital setting.

Sex, age, and medical benefits (i.e., type of insurance according to the patient’s economic condition) were selected as patient characteristics. Comorbidity refers to the 17 diseases in the Charlson Comorbidity Index Deyo method selected according to disease severity (except for AIDS). As per prior research, risk factors include mental health problems, physical illness, financial problems, and conflicts with family members [16,17]. The means of self-harm include poisoning, cutting, suffocation, and falling.

### 2.3. Data Analysis

The patient characteristics, comorbidities, risk factors, and means of self-harm of the patients who attempted self-harm in the hospital setting were indicated as frequencies and percentages, and chi-square and Fisher’s exact tests were performed. In addition, a logistic regression analysis was conducted to identify the factors affecting the performance of self-harm in the hospital setting.

Then, the patients who attempted self-harm in the hospital setting were divided into survival and death groups. The frequencies and percentages of their characteristics, risk factors, and means of self-harm were described. The chi-square test and Fisher’s exact test were used, and logistic regression analysis was performed to analyze the factors influencing death. The frequencies regarding the period from hospitalization to self-harm attempts were also calculated. A statistical analysis was performed using STATA, version 15.0 (StataCorp), and statistical significance was set at 5%. The analyzed data excluded personally identifiable information, such as medical institution code numbers and patient registration numbers.

## 3. Results

### 3.1. Participants’ Characteristics

Table 1 shows the descriptive characteristics of the study sample (N = 7235 people). There were more female (N = 4195, 58.3%) than male participants (N = 2997, 41.7%), and the mean age was 47.8 years (±19.4). In total, 8.6% of the participants received medical benefits. Regarding comorbidities, there were cases of cerebrovascular disease (1.3%), renal disease (0.8%), cancer (1.5%), and metastatic cancer (0.4%).

The number of people who performed self-harm outside the hospital setting was 7192, and that of those who performed self-harm in the hospital setting was 43 (0.59% of the sample). Among patients who attempted self-harm inside the hospital setting, there were more men (N = 24, 55.8%; women: N = 19, 44.2%), and the average age was 56.3 years (±20.45). Of these, 20.9% received medical benefits. Regarding comorbidities, there were cases of cerebrovascular disease (7.0%), renal disease (4.7%), cancer (16.3%), and metastatic cancer (7.0%). The risk factors with the highest rates were 14.0% for physical illness and 7.0% for conflict with family members, and the primary means of self-harm were poisoning (62.8%) and falling (37.2%). Death was the outcome of the treatment in 27.9% of the cases.

Regarding the risk factors with the highest rates among patients who attempted self-harm outside the hospital setting, 5.4% were for physical illness and 22.9% for conflict with family members. Poisoning (96.2%) and falling (3.5%) were used as the primary means of self-harm. The treatment outcome was death in 8.4% of the cases.

### 3.2. Factors Influencing Self-Harm in the Hospital Setting

Among the factors influencing self-harm in the hospital setting (Table 2), an increase in one unit of age led to an increase of 1.025 in the chance of performing self-harm (odds ratio [OR] = 1.025, 95% CI = 1.006–1.045), and receiving medical benefits increased this chance by 2.383 times (OR = 2.383, 95% CI = 1.051–5.404). Regarding comorbidities, having peptic ulcer disease increased the chance of performing self-harm by 4.944 times (OR = 4.944, 95% CI = 1.062–23.009), and this increase was 2.045 times for those with cancer (OR = 2.045, 95% CI = 1.081) −3.870). Having financial problems increased the chance of performing self-harm by 3.285 times (OR = 3.285, 95% CI = 0.066–10.128).

### 3.3. Treatment Outcomes of Patients Who Attempted Self-Harm in the Hospital Setting and Their Characteristics

Table 3 shows the differences in the characteristics of patients who attempted self-harm in the hospital setting by treatment outcome. There were statistically significant differences by sex, physical illness, and means of self-harm. Among the patients who attempted self-harm in the hospital setting, 31 survived, and 12 died. Among those who survived, there were more women (54.8%; men: 45.2%), the average age was 54.7 years (±21.1), only physical illness (6.5%) showed high rates among the risk factors of self-harm, and poisoning (80.6%) and falling (19.4%) were used as the means of self-harm.

Among those who died, there were more men (83.3%; women: 16.7%), the average age was 60.6 years (±18.9), physical illness (33.3%) was a significant risk factor of self-harm, and poisoning (16.7%) and falling (83.3%) were used as the means of self-harm.

### 3.4. Factors Influencing Mortality among Patients Who Attempted Self-Harm in the Hospital Setting

In this study, receiving medical benefits, which may be related to people’s financial status, and conflict with family members, were also influencing factors of the performance of self-harm in the hospital setting (Table 4). Yes indicates that female has the presence of disease. However, attempting self-harm using poisoning decreased the chance of death by 0.014 times (OR = 0.014, 95% CI = 0.001–0.304).

### 3.5. Period from Admission to the Date of the Self-Harm Attempt

The number of patients and the rates for the period from admission to the date of the self-harm attempt are shown in Table 5. The rates were the highest for attempting self-harm on the day of admission (18 patients, 41.9%) and within one week (1–7 days) (11 patients, 25.6%).

## 4. Discussion

Self-harm and suicide are serious issues not only in South Korea, but also worldwide. Therefore, we draw on similarities and references from other countries regarding the influencing factors of self-harm and suicide and how to ameliorate the public health environment to reduce their rates.

These findings broadly resemble the evidence, from various countries, found in prior research. In the United Kingdom, 36% of 55 older adult patients admitted to hospital due to acute medical conditions had suicidal thoughts, and 22% expressed a wish to die [18]. Among Taiwanese inpatients, younger female and older male patients had a higher risk of suicide attempts and suicide-related mortality, respectively [19]. Another study in Taiwan showed that 3.1% of older adult hospitalized patients with medical or surgical conditions had suicidal ideation [20].

In the current study, the most common period from the date of admission to the date of the self-harm attempt was the same day of hospitalization (18 patients, 41.9%), followed by within a week of hospitalization. The data on these periods can be used as indicators to classify high-risk groups of self-harm and suicide, and to develop guidelines and policies to strengthen patient safety. The results show that healthcare professionals should pay special attention to high-risk patients for suicide and self-harm during the first day of hospitalization.

In this study, the primary means of self-harm in the hospital were poisoning and falling. Further, self-harm by poisoning reduced the chance of death by suicide by 0.014 times. These results are similar to those reported in other countries. In Taiwan, an analysis of national data collected between 2013 and 2019 by the National Health Security Office and the National Death Certification Registry System showed that drug poisoning was the most frequently used method (45.34%) of self-harm attempts, followed by pesticide poisoning (26.55%) [19]. In Helsinki, Finland, between 1963 and 2000, researchers accompanied 98 out of 100 suicide attempters in a drug-addiction ward, and their results showed that 13 patients (6 women out of a total of 71; 7 men out of a total of 27) died by poisoning (62%), 8% by hanging, and 8% by jumping. Moreover, 8 (62%) of the 13 patients died within 15 years, and the mortality rate was high [21]. Regarding our results on the means of self-harm and suicide, we see that the need to discuss them with the results from prior research in other countries as the means of self-harm and suicide (and their related countermeasures within hospital settings) in international settings may be directly related to inpatients’ suicide-related status. The related analyses may help invested stakeholders devise optimized recommendations for patient safety in Korea. For example, hanging was, by far, the most common method of inpatient suicide according to the National Violent Death Reporting System (33 of 46, 71.7%) and the Joint Commission’s Sentinel Event databases (137 of 195, 70.3%) [8]. Further, the Joint Commission’s Sentinel Event database showed that doors, door handles, or fixed-joint points (e.g., hinges) were used for 53.8% of the hanging methods in 106 of the 137 hospitalized patients. Hence, they devised three sentinel event alerts to provide hospitals and other healthcare organizations with guidance on preventing suicide inside their institutions. These and other prevention recommendations have focused on conducting risk assessments, improving environmental safety (e.g., removing ligature points), and implementing risk-mitigation strategies (e.g., protective observation policies and procedures) [22]. These data may help advance patient safety guidelines in Korea using a multi-faceted approach tailored to the country’s situation.

In this study, the factors influencing self-harm in general hospitals in Korea were related to comorbidities. Specifically, the highest rates regarding comorbidities were for cerebrovascular disease and cancer. Furthermore, those with peptic ulcer disease and cancer were 4.944 and 2.045 times more likely to perform self-harm in the hospital setting, respectively. Once more, our results show consistency in the findings of prior international research. A study that examined the prevalence and correlation of suicidal ideation among Chinese inpatients with cancer in large general hospitals reported that the risk of suicide was the highest among patients with non-localized cancer or a poor prognosis [23]. In another study, late-stage cancer was more lethal and less treatable than early-stage cancer, and patients with terminal cancer had the highest level of hopelessness, which is a powerful predictor of suicidality in cancer patients [24].

In Korea, the government provides medical insurance and public assistance programs for people with financial disadvantages, including providing financial support for people to take care of their primary needs, such as food and clothing. In our data, 20.9% of participants received medical benefits, and the risk factor with the highest rate in this group was a conflict with family members (7.0%); specifically, receiving medical benefits increased the chance of performing self-harm in the hospital by 2.383 times, and having financial problems increased this chance by 3.285 times. These results are very similar to studies in other countries, which show that economic factors influence suicide in hospital settings. Scholars examining the reports of the side effects of inpatient suicide on people working in the hospital between 2008 and 2014 in China interviewed six medical staff members who experienced inpatient suicide [25]; according to the influencing factors of suicide for hospitalized patients, these patients were classified into five high-risk groups of suicide. Two of these groups were related to economic difficulties, such as funding for medical expenses and social support to cover medical expenses. Specifically, when patients with mental stress due to diseases were required to pay medical bills without social support, they feared that they would burden their families with these costs, leading to suicidal thoughts and suicide [25]. In addition, the study participants who did not have social support tended to lose faith in their lives, and 11 died by suicide due to a lack of social support, which could expose the low quality of the health environment or a lack of will to live [25].

In Taiwan, academicians analyzed data on medical claims from inpatients and outpatients between 1998 and 2015 from the Health and Welfare Data Center of the Ministry of Health and Welfare, discovering a high rate of death for low-income inpatients caused by intentional injuries (i.e., suicide and self-harm) [26]. The likelihood of performing such injuries was also 2.014 times higher in low-income patients. The mortality rates of low-income patients with intentional and unintentional injuries (31% vs. 1.7%, respectively) were also higher than those of non-low-income patients (4.4% vs. 3.1%, respectively) [26]. The researchers then described that low-income patients could experience hopelessness very quickly and that higher mortality rates due to “medical negligence” can lead to health inequality. In another study, the mortality rate of low-income inpatients was 1.888 times higher than that of non-low-income inpatients, again revealing health inequality related to economic factors [27].

## 5. Limitations

First, the data set we used pertains to discharged patients who intentionally performed self-harm and suicide attempts. One disadvantage of this data set lies in its non-inclusion of data from patients who did not visit the hospital or were not hospitalized because of death or discharge in the emergency ward. Another disadvantage is that the data set was limited, potentially indicating a restriction in the sample range. Despite these limitations, the data we analyzed pertained to patients from medical institutions with more than 100 beds, and the data set was not limited to specific hospitals. Thus, the data may be generalizable at the national level according to survival and death by self-harm.

Second, other factors influencing suicide, suicide attempts, and self-harm in the hospital setting were not considered. For example, this study did not examine psychological factors such as depression, suicide patterns, personality disorders, major life events, and stress. Still, we hypothesize that these factors can interact to reveal unique suicide attempt characteristics. Future studies may include expanded research that reflects on the complex factors influencing in-hospital suicide and self-harm.

Finally, the data set we used is based on voluntary reports, highlighting the possibility of various limitations in the amount of data collected. Research shows that there is not only a general reluctance to report deaths by suicide due to societal stigma, but also inadequacies in death registration practices, with incomplete or erroneous entries for the cause of death occurring in official records [19]. These limitations could be overcome if the relevant bodies provide recommendations for medical institutions to report on suicide deaths and attempts, so that the relevant data can be valuably applied to improve social awareness about suicide reporting in hospitals and build a suicide prevention plan.

## 6. Conclusions

To curtail the suicide rates among patients in general hospitals in Korea, it may be necessary to strengthen patient safety in these institutions. This can be operationalized by measures aimed at reducing the number of suicide attempts, access to means of self-harm, and other measures. The evidence in this study can assist in establishing relevant policies aimed at reducing mortality rates related to in-hospital suicide. This study advances the literature by providing data that distinguishes the place wherein the person attempted self-harm (i.e., in and outside the hospital setting) and that identifies and typifies high-risk patients among those who perform self-harm. The study also demonstrates that it would be wise for invested stakeholders to develop interventions and policies that tackle the effects of comorbidities and economic disadvantages on self-harm; they should also pay attention to older adult patients and male patients, as they showed higher self-harm rates. Our results further show that the primary means of self-harm in the hospital were falling and poisoning, that the subsequent mortality for these cases was high, and that the rate of self-harm attempts within a short period after hospitalization was high (See Table 5). Since the risk factors analyzed in this study differ by patients’ characteristics, researchers should conduct studies examining the causative variables of the risk factors among patients at high risk for self-harm. Our study results also identify the characteristics and influencing factors of inpatient self-harm, allowing invested stakeholders to develop a model for predicting patients at a high risk of self-harm and create preventive policies to reduce the risk of self-harm among inpatients, for example, through conducting appropriate risk assessment activities. Specifically, this study delivers data demonstrating the high-risk factors for self-harm attempts, the characteristics of patients who attempted self-harm in the hospital setting, the influencing factors of self-harm, and the characteristics of patients who attempted self-harm outside of the hospital setting. We used data from the Korean National Hospital Discharge In-depth Injury Survey to contribute relevant evidence for strengthening patient safety in hospitals domestically.

## Figures and Tables

**Table 1 ijerph-20-03095-t001:** Participants’ characteristics by the place where patients performed self-harm.

Characteristics		Division	Outside Hospital		Inside Hospital		χ^2^*/t*	*p*
			Means, Standard Deviation, and Frequency	(%)	Means, Standard Deviation, and Frequency	(%)		
All			7192		43			
Characteristics of patients	Sex	Male	2997	41.7	24	55.8	3.515	0.061
		Female	4195	58.3	19	44.2		
	Age (mean) SD		47.8	19.4	56.3	20.4	−2.880	0.004
	Medical benefit		620	8.6	9	20.9	8.160	0.004
Comorbidities	AMI		23	0.3	1	2.3	5.201	0.134
	CHF		48	0.7	1	2.3	1.747	0.254
	PVD		11	0.2	0	0.0	0.660	1.000
	CEVD		92	1.3	3	7.0	10.708	0.019
	Dementia		85	1.2	2	4.7	4.331	0.094
	COPD		97	1.3	1	2.3	0.305	0.445
	Rheumatoid disease		10	0.1	0	0.0	0.060	1.000
	PUD		75	1.0	2	4.7	5.286	0.076
	Mild LD		174	2.4	2	4.7	0.897	0.281
	Diabetes		386	5.4	4	9.3	1.298	0.292
	Diabetes complications		31	0.4	0	0.0	0.186	1.000
	HP/PAPL		28	0.4	0	0.0	0.168	1.000
	RD		54	0.8	2	4.7	8.467	0.043
	Cancer		107	1.5	7	16.3	60.301	0.000
	Moderate/severe LD		16	0.2	0	0.0	0.096	1.000
	Metastatic cancer		26	0.4	3	7.0	46.855	0.001
Risk Factor	Mental problems		1958	27.2	15	34.9	1.264	0.261
	Physical illness		385	5.4	6	14.0	6.184	0.013
	Financial problems		408	5.7	5	11.6	2.816	0.096
	Conflict with family members		1649	22.9	3	7.0	6.173	0.010
Means of self-harm	Poisoning		6917	96.2	27	62.8	123.416	0.000
	Cutting		8	0.1	0	0.0	0.048	1.000
	Hanging		3	0.0	0	0.0	0.018	1.000
	Falling		249	3.5	16	37.2	137.960	0.000
Result		Death	602	8.4	12	27.9	21.007	0.000

AMI: acute myocardial infarction, CHF: congestive heart failure, PVD: peripheral vascular disease, CEVD: cerebrovascular disease, COPD: chronic obstructive pulmonary disease, PUD: peptic ulcer disease, LD: liver disease, HP/PAPL: hemiplegia or paraplegia, RD: renal disease.

**Table 2 ijerph-20-03095-t002:** Factors influencing self-harm in the hospital setting.

		OR	*p*	95% CI
Sex (Male)		1.002	0.994	0.513–1.959
Age		1.025	0.011	1.006–1.045
Medical benefit (no)	Yes	2.383	0.038	1.051–5.404
AMI (no)	Yes	5.425	0.136	0.588–50.041
CHF (no)	Yes	3.034	0.315	0.349–26.412
PVD (no)	Yes	1.000		
CEVD (no)	Yes	3.606	0.069	0.907–14.334
Dementia (no)	Yes	1.498	0.657	0.251–8.947
COPD (no)	Yes	1.002	0.998	0.129–7.818
Rheumatoid disease (no)	Yes	1.000		
PUD (no)	Yes	4.944	0.042	1.062–23.009
Mild LD (no)	Yes	1.373	0.697	0.278–6.771
Diabetes (no)	Yes	0.868	0.810	0.273–2.759
Diabetes complications (no)	Yes	1.000		
HP/PAPL (no)	Yes	1.000		
RD (no)	Yes	1.551	0.330	0.641–3.748
Cancer (no)	Yes	2.045	0.028	1.081–3.870
Moderate/severe LD (no)	Yes	1.000		
Metastatic cancer (no)	Yes	1.116	0.489	0.818–1.521
Mental problem (no)	Yes	1.660	0.210	0.751–3.669
Physical illness (no)	Yes	1.652	0.381	0.538–5.076
Financial problem (no)	Yes	3.285	0.038	1.066–10.128
Conflict with family members (no)	Yes	0.634	0.488	0.175–2.294
Poisoning (no)	Yes	13,689.900	0.988	
Cutting (no)	Yes	1.000		
Hanging (no)	Yes	1.000		
Falling (no)	Yes	277,593.600	0.984	

AMI: acute myocardial infarction, CHF: congestive heart failure, PVD: peripheral vascular disease, CEVD: cerebrovascular disease, COPD: chronic obstructive pulmonary disease, PUD: peptic ulcer disease, LD: liver disease, HP/PAPL: hemiplegia or paraplegia, RD: renal disease.

**Table 3 ijerph-20-03095-t003:** Characteristics of patients who attempted self-harm in the hospital setting by treatment outcome.

Characteristics		Division	Survival		Mortality		χ_2_/*t*	*p*
			Frequency	(%), Mean, Standard Deviation	Frequency	(%), Mean, Standard Deviation		
All			31		12			
Characteristics of patients	Sex	Male	14	45.2	10	83.3	5.111	0.039
		Female	17	54.8	2	16.7		
	Age		54.7	21.1	60.6	18.9	−0.847	0.402
	Medical benefit		6	19.4	3	25.0	0.167	0.692
Risk Factor	Mental problem		13	41.9	2	16.7	2.432	0.164
	Physical illness		2	6.5	4	33.3	5.207	0.042
	Financial problem		4	12.9	1	8.3	0.176	1.000
	Conflict with family members		3	9.7	0	-	1.248	0.548
Mean of self-harm	Poisoning		25	80.6	2	16.7	15.156	0.000
	Falling		6	19.4	10	83.3	15.156	0.000

**Table 4 ijerph-20-03095-t004:** Factors affecting the mortality of patients who performed self-harm in the hospital setting.

		OR	*p*	95% CI
Sex (male)	Female	0.202	0.315	0.009–4.565
Age		1.016	0.723	0.933–1.106
Medical benefit (no)	Yes	0.835	0.896	0.056–12.461
Mental problem (no)	Yes	0.152	0.321	0.004–6.284
Physical illness (no)	Yes	4.228	0.403	0.144–124.146
Financial problem (no)	Yes	0.097	0.237	0.002–4.626
Conflict with family members (no)	Yes	1.000		
Poisoning (no)	Yes	0.014	0.007	0.001–0.304
Falling (no)	Yes	1.000		

**Table 5 ijerph-20-03095-t005:** The period from admission to the date of the self-harm attempt.

Period	Number of Patients	(%)
Pre-hospitalization ^1^	7	16.3
The day of admission	18	41.9
Within one week (1–7 days)	11	25.6
Within two weeks (8–14 days)	2	4.7
Within three weeks (15–21 days)	2	4.7
More than four weeks (after 22 days)	3	7.0
Total	43	100.0

^1^ In the case of pre-hospitalization, the patient was transferred to the hospital, and the date of occurrence cannot be determined.

## Data Availability

The data that support the findings of this study are available from https://www.kdca.go.kr/injury/ (accessed on 25 May 2022).

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
