# Peer review of "Factors Affecting Inpatients’ Mortality through Intentional Self-Harm at In-Hospitals in South Korea"

_ijerph, 2023, doi:10.3390/ijerph20043095_

Round 1
Reviewer 1 Report (Previous Reviewer 2)
The authors have improved the paper and addressed some of my comments. However, some of them remain:
3. If possible, provide the source (i.e., URL) of the data set.
Response: It is not possible to provide it.
Reviewer' reply: So, provide the reason for this decision in the Data Availability Statement section.
4. Table 2 should have columns explained in the text. I could not understand the column with values ' Yes'. Idem for Table 4.
Response: Yes meaning is that it has a disease or problem and compare with not have via statistical analysis.
Reviewer' reply: Explain this point in the text.
7. Finally, I think the content from line 336 to 358 should be in a subsection, and it must be expanded to detail:
a) How you "hope the results can be used as fundamental data for developing programs that complement and strengthen policies and monitoring systems related to patient safety in medical institutions";
b) How "they can also be used for developing appropriate measures to reduce the number of hospital suicides";
c) How you "hope that YOUR evidence assists in reducing hospital suicide rates".
Response: From 335 to 358 can be "result" or "conclusion" after limitation or discussion
Reviewer' reply: Remove references from the conclusion. There should not be references in there. If you want to use them, the content should be in the discussion section, not in a conclusion section.
Author Response
Please see the attachment.

This manuscript is a resubmission of an earlier submission. The following is a list of the peer review reports and author responses from that submission.
Round 1
Reviewer 1 Report
This paper uses existing medical data to examine factors around self-harm and suicide in people admitted to hospital. While the results are fairly modest due to limited data being included, this is a fairly novel topic and may have benefits for patient safety. Some changes need to be made to bring this up to publishable standard.
1. the topic, rationale and aims of the study are too vague and seem to change throughout the paper. The main topic is about describing outcomes following self-harm among people who are admitted to hospital. However, results for the comparison group of people who self-harm outside the hospital are given too much emphasis at times. These data are useful for comparison and to add context to the admitted patient group, but reporting on them can be distracting. It needs to be made much clearer from the start, which group you are looking at, and why. The rationale needs to be more focused around this as well e.g. why is this an important topic, what is not known about this topic and how can this study potentially help increase knowledge and/or improve care?
2. There is a lack of information in the methods section, and this needs to be improved. Some information is present in other parts of the paper and should be moved to the methods - details are provided below with line numbers. Why was only a sample used in this study rather than the full cohort? please justify the sample and how it was selected. Please define where this data comes from in more detail - is this general hospitals only or does it include psychiatric hospitals? Some of the terminology re 'medical benefits' may not be common internationally, please describe what this means e.g. lower economic status. What does voluntary data mean? Is the data collected by the hospitals? Directly from the patients? Do patients give consent for it to be shared?
3. Please also clarify your definitions of self-harm used in the study, and reference them. It was not clear if self-harm via cutting was included or not.
4. More explanation needs to be included about why comorbidities are important. As an inpatient sample these are likely to be people at the more seriously ill side of things already, was there any way to account for severity of illness, as this is likely to be a contributing factor to self-harm/suicide.
5. Can you clarify what these people died from? are you looking only at suicide deaths, or are any deaths at all included? Again inpatients are more likely to be seriously ill and therefore more likely to go on to die of natural causes.
6. The difference between methods poisoning/falling is not a main result as it is likely to be based on availability of access to medications in the hospital setting.
7. discussion should be more concise.
specific points by line number:
49-51 the quote used mentioned NHS - which is a UK institution, but is attributed to the US. Please clarify.
55-62 this section about suicidal thoughts is not needed, as the paper is not on suicidal thoughts.
82-86 should be in Methods section.
123 the 'characteristics' included might be better characterised as demographics as they are very basic.
129-131 this sentence is not needed.
229-230 this is for the results section, not discussion.
238-273 this whole section reads more like an introduction than a discussion. Please edit down. It only needs to raise points of difference and similarity with previous research - with the aim of showing this work has contributed something novel. Also the referenced work covers different settings and may not be completely comparable to the current study - broad points would be better.
304-307 should be in Methods section.
322 please use 'died by suicide' as the term 'committed suicide' is not longer used.
364 not sure what 'limited to damaged data' means
Best of luck with your revisions.
Reviewer 2 Report
This is a data analysis study focused on identifying the relevant features of patients who attempted self-harm in and outside the hospital in South Korea.
1. Check https://www.mdpi.com/journal/ijerph/instructions - "The abstract should be a total of about 200 words maximum". There are texts repeated.
2. Consider "... and 12 died, with the primary means of self-harm being 2 patients of poisoning and 10 patients of falling". In this sense, are you talking about suicide attempt? Can 'Intentional Self-Harm' be suicide attempt or behavior in some (maybe most) contexts? Text should be clearer in the abstract.
3. If possible, provide the source (i.e., URL) of the data set.
4. Table 2 should have columns explained in the text. I could not understand the column with values 'Yes'. Idem for Table 4.
5. It is weird to end a paper with a limitations section. My recommendation is to add a conclusion section to show if and how the objective of the study was archived.
6. Also, add future works in the conclusion section.
7. Finally, I think the content from line 336 to 358 should be in a subsection, and it must be expanded to detail:
a) How you "hope the results can be used as fundamental data for developing programs that complement and strengthen policies and monitoring systems related to patient safety in medical institutions";
b) How "they can also be used for developing appropriate measures to reduce the number of hospital suicides";
c) How you "hope that YOUR evidence assists in reducing hospital suicide rates".
Specific points:
8. Read and rephrase "Among 31 of the patients who performed self-harm in the hospital survived, and 12 died." - perhaps removing 'among'.
9. Check "it is estimated that between 48.5 and 64.9 hospital inpatient suicides occur per year in the United States of America", are these numbers percentages?
10. line 88: "...Therefore, this study aimed..." - the word "therefore" does not make sense in this sentence.